# Pre-Trained Multi-Goal Transformers with Prompt Optimization for Efficient Online Adaptation

Haoqi Yuan[1]    Yuhui Fu[1]    Feiyang Xie[2]    Zongqing Lu[1,3]*

[1]School of Computer Science, Peking University
[2]Yuanpei College, Peking University
[3]Beijing Academy of Artificial Intelligence

## Abstract

Efficiently solving unseen tasks remains a challenge in reinforcement learning (RL), especially for long-horizon tasks composed of multiple subtasks. Pre-training policies from task-agnostic datasets has emerged as a promising approach, yet existing methods still necessitate substantial interactions via RL to learn new tasks. We introduce MGPO, a method that leverages the power of Transformer-based policies to model sequences of goals, enabling efficient online adaptation through prompt optimization. In its pre-training phase, MGPO utilizes hindsight multi-goal relabeling and behavior cloning. This combination equips the policy to model diverse long-horizon behaviors that align with varying goal sequences. During online adaptation, the goal sequence, conceptualized as a prompt, is optimized to improve task performance. We adopt a multi-armed bandit framework for this process, enhancing prompt selection based on the returns from online trajectories. Our experiments across various environments demonstrate that MGPO holds substantial advantages in sample efficiency, online adaptation performance, robustness, and interpretability compared with existing methods.

## 1   Introduction

In the evolving landscape of deep learning, the paradigm of pre-training followed by fine-tuning has become a dominant approach for improving learning downstream tasks, particularly in the domain of computer vision [20, 62, 41] and natural language processing (NLP) [30, 32]. This paradigm has recently been explored in deep reinforcement learning (RL) [3, 4], addressing the issue of sample efficiency in RL when solving unseen tasks by leveraging the acquired knowledge during pre-training. For example, offline meta-RL (OMRL) [33, 59, 56] studies pre-training a policy on *multi-task datasets*, which adapts to a new task with limited interactions in this task. However, these approaches typically require extensive data collection for each specific task. In contrast, given the relative ease of acquiring large, task-agnostic datasets that contain diverse behaviors, other studies focus on pre-training policies [4, 54] and skills [38, 47] on *task-agnostic datasets* and adapt to unseen tasks with RL finetuning.

Most works [56, 55], especially in the context of OMRL, primarily focus on short-term tasks with shaped rewards, where information of the unknown task can be inferred within a few steps of online interactions (e.g., MuJoCo [50] and MetaWorld [58]). These approaches often fall short of tackling long-horizon tasks which are characterized by a sequence of sub-processes. In our study, we consider this challenging yet realistic scenario: given a task-agnostic dataset characterized by diverse behaviors of the agent, we aim to pre-train a policy to facilitate efficient online adaptation to new, long-horizon tasks.

---

*Correspondence to Zongqing Lu <zongqing.lu@pku.edu.cn>.

38th Conference on Neural Information Processing Systems (NeurIPS 2024).

Recent studies [38, 60] tackle this problem by assuming that diverse short-term skills can be acquired from task-agnostic datasets, which are sufficient to compose the required behaviors for complex, long-horizon tasks. These methods pre-train a goal-conditioned policy focused on short-term skills. During online adaptation, they use RL to train a separate high-level policy that selects goals based on current states and executes the pre-trained policy for multiple steps. While this approach avoids the extensive finetuning of the pre-trained policy, it still requires substantial number of interaction steps for online RL. This limitation arises from the pre-trained policy's inability to sequentially achieve multiple goals during a single attempt. The capability to switch goals – deciding at which state to transition to the next goal – must be developed during the costly online adaptation phase.

*Is it feasible to pre-train a policy conditioned on a sequence of goals, capable of autonomously transitioning between goals while executing long-horizon behaviors aligned with these goals?* Achieving such a policy would pave the way for developing efficient online adaptation algorithms, eliminating the need for learning a high-level policy for state-conditioned goal switching. However, the challenge lies in the substantial demand on the policy's long-term memory capabilities, which must effectively remember the goal sequence and base action prediction on a long history context.

To tackle this challenge, we draw inspiration from the success of Transformers [51] in language modeling [10, 7], which have demonstrated a remarkable capability to model long sequences and, when pre-trained on diverse datasets, adapt efficiently to downstream tasks through prompt optimization [25]. In our proposed Multi-Goal Transformers with Prompt Optimization (**MGPO**), we pre-train a Transformer-based policy that takes a goal sequence as a prompt and predicts actions in the following sequence of environment observations. During online adaptation, we only optimize for the sequence of goals in the prompt, drawing parallels with the concept of prompt optimization in language models.

In the pre-training stage, we employ hindsight relabeling to construct prompts from goals visited in the trajectory and train the Transformer policy through behavior cloning. Thus, the policy learns to produce behaviors consistent with the goal sequence. The number of goals in the prompt modulates the deterministic or exploratory nature of the policy's behavior. For online adaptation, we propose a novel prompt optimization strategy, leveraging the returns of online trajectories to guide the selection of goal sequences. This process is formulated as a multi-armed bandit problem and we introduce two approaches through upper confidence bound (UCB) and Bayesian posterior estimation. Figure 1 provides an overview of our framework.

We evaluate MGPO across diverse domains, including maze navigation, the robotic simulation environment Kitchen, and the open-world game Crafter. Our results demonstrate that MGPO significantly surpasses prior methods in terms of sample efficiency and performance during online adaptation. MGPO adapts to new tasks within a small budget of 100 online episodes in all environments. Comparisons to existing prompt optimization methods highlight the interpretability and robustness of our method.

In summary, our main contributions are:

- We propose pre-training multi-goal Transformers to address the challenge of online adaptation in long-horizon tasks. Our approach combines the strengths of Transformers in sequence modeling and the advantages of goal-conditioned pre-training for task adaptation.

- We introduce novel methods for optimizing goal sequences, offering enhanced interpretability and robustness compared to existing prompt optimization methods.

- Our experimental results in diverse, challenging environments demonstrate that MGPO significantly enhances sample efficiency over existing methods.

## 2 Preliminaries

### 2.1 Problem Formulation

A task in an environment is formulated as a partially observable Markov Decision Process (POMDP) $M = \langle S, O, A, T, \rho, r, \gamma \rangle$ representing the state space, the agent's observation space, the action space, the transition probability of the environment, the initial state distribution, the reward function, and the discount factor, respectively. Starting from the initial state, the agent takes an action at each timestep, and then the environment transitions to the next state and returns a reward. At each timestep

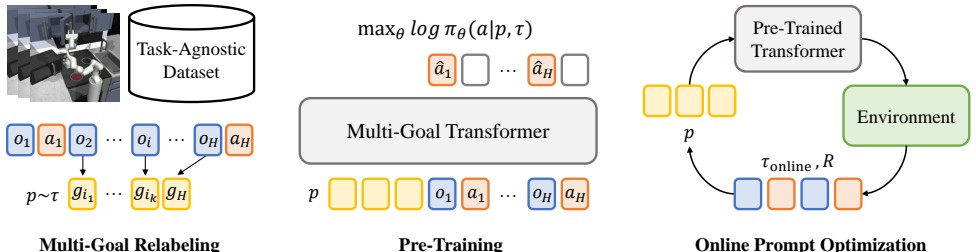

**Figure 1:** An overview of MGPO. We pre-train Transformer-based policies on task-agnostic datasets, leveraging hindsight multi-goal relabeling and behavior cloning to endow policies with the capacity for modeling long-term behaviors. In unseen tasks, multi-goal Transformers adapt efficiently through prompt optimization, which searches for a sequence of goals with the aim of maximizing online returns.

$t$, the environment provides the agent an observation $o_t$ via an emission function $o_t \sim E(o_t|s_t)$. This process continues until the task either terminates or reaches a maximum timestep $H$. We denote the historical observations and actions as $h_{t-1} = \{(o_i, a_i)\}_{i=1}^{t-1}$. Reinforcement learning (RL) aims to learn a policy $\pi(a_t|h_{t-1}, o_t)$ to maximize the expected return $J(\pi) = \mathbb{E}_{a_t \sim \pi(a_t|h_{t-1}, o_t)} [R]$, where $R = \sum_{t=1}^{H} \gamma^t r(s_t, a_t)$ is the discounted cumulative return of an episode.

We focus on long-horizon tasks that require executing a substantial number of subtasks sequentially. We define a goal function $g = f_G(o)$ that maps an observation to a goal, representing key information related to task completion, such as the agent's position in navigation tasks or the states of objects in robotic scenarios. $G$ denotes the goal space. A goal $g$ is said to be reached at time $t$ if $g_t = f_G(o_t)$. We assume that the task $M$ in an unknown environment provides a task goal $g^M$, indicating the final goal to be reached. To optimally solve the long-horizon task, the policy must sequentially reach multiple goals. As an example, consider an environment resembling rooms with unknown structures where the task is to navigate to a goal location $g^M$. Under partial egocentric observations, the agent should learn from trial and error to find the shortest path to the goal location, which involves reaching several specific waypoints $(g_1, ..., g_k)$ as necessary goals.

During pre-training, we assume access to a task-agnostic dataset $D = \{\tau_i\}_{i=1}^{N}$ consisting of trajectories $\tau = (o_1, a_1, ..., o_H, a_H)$, in which the agent performs diverse behaviors. For online adaptation, we aim to solve an unseen task $M$ with its goal $g^M$ provided, within an online interaction budget of $N$ episodes. The objective is to find the optimal policy $\pi$ maximizing the expected return $J(\pi)$.

## 2.2 Transformers and Prompt Optimization

Some previous methods [4] pre-train a policy $\pi_\theta(a_t|h_{t-1}, o_t)$ parameterized by $\theta$ and then finetune with online RL. This approach is sample-inefficient due to discrepancies between the behaviors in the dataset and those required for downstream tasks. Other methods [38, 60] pre-train a policy $\pi_\theta(a_t|z, h_{t-1}, o_t)$ conditioned on a variable $z$ to perform diverse short-term skills and then train a high-level policy $\pi_\phi(z|h_t, o_t)$ online using RL. As discussed earlier, this approach often results in limited sample efficiency because the high-level policy must learn when to switch skills $z$ based on the observations during the online phase.

In this work, we explore pre-training a policy capable of performing diverse long-horizon behaviors, aiming to develop efficient online adaptation methods that do not require additional RL. To model such diverse long trajectories, we employ the Transformer architecture in our policy, known for its effectiveness in sequence modeling tasks. Transformers leverage an attention mechanism [51], enabling the model to weigh different parts of the input sequence differently, thus effectively addressing long-term dependencies.

In pre-training, we aim to develop a Transformer-based policy $\pi_\theta(a_t|p, h_{t-1}, o_t)$ parameterized by $\theta$, taking an additional input variable $p$. This policy processes inputs in an auto-regressive manner, where the input is a sequence $(p, o_1, a_1, ..., o_H, a_H)$, and outputs an action $a_t$ at each observation $o_t$. Here, the input variable $p$ acts similarly to a prompt in language models, guiding the policy's behavior. During online adaptation, we keep the model parameters $\theta$ fixed and treat $p$ as the optimization

variable, transforming the problem into finding an optimal prompt $p$:

$$\max_p J\left(\pi_\theta\left(a_t|p, h_{t-1}, o_t\right)\right). \tag{1}$$

This approach mirrors the concept of prompt optimization in language models and eliminates the need to use online RL to train a high-level policy.

## 3 Method

### 3.1 Pre-Training Multi-Goal Transformers

We implement a prompt as a sequence of goals, describing an ordered sequence of key states to be visited, which abstracts the long-term behavior of the agent. Inspired by recent works [28, 60] that employ hindsight relabeling [1] to generate goals from offline trajectories, we sample a sequence of goals within the trajectory to construct the prompt.

Using the observations $(o_1, ..., o_H)$ in a trajectory $\tau$, we construct a sequence of goals $g = (g_1, ..., g_H)$ with the same length $H$, representing the process of agent behavior in this episode. We first uniformly sample a number $k \sim U[0, K-1]$ where $K$ is the maximal prompt length, then uniformly sub-sample the goal sequence to construct a prompt $p = (g_{i_1}, ..., g_{i_k}, g_H)$, where $0 \le i_1 \le ... \le i_k < H$. Since the trajectory ends at the last goal $g_H$, we keep $g_H$ at the last position in the prompt. Using this sampling mechanism, the prompt can describe behavior in different granularities with its varying length. We denote the process of sampling a prompt from the trajectory as $p \sim P(p|\tau)$.

Similar to Decision Transformers [8], we adopt a causal Transformer to build the policy $\pi_\theta(a_t|p, h_{t-1}, o_t)$, which takes as input a sequence concatenating the prompt and the trajectory: $(p, \tau) = (g_{i_1}, ..., g_{i_k}, g_H, o_1, a_1, ..., o_H, a_H)$. At each input $o_t$, the policy only sees the sub-sequence from $g_{i_1}$ to $o_t$ due to the causal attention mask and predicts the action distribution $\pi_\theta(a_t|p, h_{t-1}, o_t)$. We sample batches of trajectories in $D$ and train the policy using behavior cloning:

$$\max_\theta \mathbb{E}_{\tau \in D, \, p \sim P(p|\tau)} \left[\sum_{t=1}^{H} \log \pi_\theta(a_t|p, h_{t-1}, o_t)\right]. \tag{2}$$

This training scheme encourages the policy to utilize the information provided in the prompt to reduce uncertainty in action prediction. After pre-training, it learns to perform behaviors that follow the sequence of goals in the prompt and seamlessly alter behavior between different goals. Prompts with different numbers of goals provide varying amounts of information to match behavior [14], thereby introducing different levels of uncertainty in action, enabling a trade-off between exploration and exploitation in online adaptation.

AMAGO [15] also adopts a similar multi-goal relabeling strategy to train RL policies. While it utilizes hindsight relabeling for better exploration, we are the first to use this scheme to pre-train policies capable of stitching different goals sequentially, thereby facilitating efficient task adaptation.

### 3.2 Online Prompt Optimization

Given an unseen task with the task goal $g^M$, we start with an initial prompt $p_0 = (g^M)$ and aim to find an optimal prompt $p^*$ with a maximal length of $K$ that maximizes the expected return. The algorithm can evaluate prompts for $N$ episodes to return an optimized prompt. In principle, we can employ any black-box optimization approach for this purpose, including discrete prompt search [9, 40] and continuous prompt-tuning [48, 27] methods. However, prompts optimized in the entire prompt space $G^K$ or even in a continuous space can contain uninterpretable goals and are out of training distribution. The policy with such prompts may produce unpredictable behaviors.

To enhance interpretability and robustness, we propose a novel method that samples prompts from online collected trajectories, aligning more closely with the training distribution of prompts. We assume that, if we condition the policy on prompts sampled from a trajectory with high return, it is likely to yield high expected returns since it performs similar behavior of this trajectory. Thus, we propose a method that alternates between exploring online trajectories for high returns and sampling new prompts from the best seen trajectory of the highest return.

Formally, we maintain a buffer $B$ of sampled prompts and their history of returns and the trajectory $\tau^*$ of the highest return $R^*$. Each iteration involves: (1) selecting a prompt $p^*$ from $B$ most likely to yield high returns and collecting a trajectory with $\pi(a|p^*, h, o)$; (2) sampling a new prompt $p' \sim P(p|\tau^*)$, which is then added to the buffer with the return obtained with $\pi(a|p', h, o)$. Prompt selection in (1), which requires acting under uncertainty given observed returns in history, can be modeled as a multi-arm bandit (MAB) problem. We implement two optional solutions including upper-confidence bound (UCB) [2] and a method based on Bayesian posterior estimation (BPE) [36], where the former selects the prompt with the highest UCB of expected return and the latter selects the prompt most possible to yield return higher than $R^*$ based on the estimated posterior of its return distribution.

In summary, our method employs several key designs to optimize prompts effectively: (1) Trajectory-based sampling: Instead of exploring the combinatorial space of goal sequences, we restrict our search to prompts derived from collected trajectories, ensuring both feasibility and relevance. (2) Reward-guided exploration: We further refine our prompt search by selecting prompts from trajectories that have highest returns, thereby enhancing the likelihood of performance improvement. (3) Task-goal consistency: We maintain the final goal within each prompt as the task goal, ensuring that all exploration efforts are aligned with task completion.

Our prompt optimization method is detailed in Algorithm 1 in Appendix E.1, where the implementations of UCB and BPE are also provided.

# 4 Experiments

In this section, we present experimental results obtained across various domains to evaluate the efficacy of MGPO. We aim to answer three questions: (1) Does MGPO improve sample efficiency in solving new tasks compared to previous methods? (2) How does our proposed prompt optimization method compare with existing methods? (3) How does each component in MGPO contribute to efficient online adaptation?

## 4.1 Environments and Datasets

Our evaluation spans multiple domains featuring long-horizon tasks. We collect datasets to pre-train models and evaluate their online adaptation capabilities on test sets of unseen tasks or environment configurations. Detailed descriptions of these environments and datasets are available in Appendix C.

**MazeRunner:** A 2D Maze navigation environment with partial observation as introduced in [15]. The maze has randomly generated walls, where the task is to reach a designated goal position $(x, y)$. The agent's observation includes its position and nearby terrain, receiving a reward of +1 for reaching the goal and a -0.1 penalty for each timestep. We collect the dataset with a handcrafted policy that explores various goals within each trajectory. Online adaptation requires finding optimal paths in unknown mazes with limited trials. We test MazeRunner with two maze sizes, $15 \times 15$ and $30 \times 30$, where the latter has a longer horizon of 500 steps.

**Kitchen and GridWorld:** Kitchen [16] is a robotic environment with continuous observations and actions, where a 7-DoF robot arm manipulates diverse objects in a simulated kitchen scene. We define each long-horizon task as completing a sequence of $n$ subtasks in a specific order, providing $+\frac{1}{n}$ reward when the next correct subtask is completed. The task goal is provided as a set of subtasks without revealing their order. Other goals are represented as one-hot vectors indicating the next subtask. We collect the dataset using policies trained with RL. We also introduce GridWorld, a simplified 2D version of Kitchen, where the agent navigates to switch states of 7 objects located in different positions in specified orders.

**Crafter** [18]**:** A simplified benchmark of the open-world game Minecraft, where the 2D world is procedurally generated. The agent receives $64 \times 64$ egocentric image observations and takes discrete actions. The objective is to unlock 22 achievements, each providing a reward of +1. Goals are defined as one-hot vectors indicating the next achievement. The dataset is collected using policies from AD [23]. For online adaptation, each task features a unique, unexplored world map.

**Table 1:** Performance of MGPO compared with baseline methods. Each result shows the average performance on all test tasks in the environment and the standard deviation across 3 random seeds for online test. Goal-conditioned BC has no error bars since it does not perform online optimization.

| Method | MazeRunner-15 | MazeRunner-30 | Kitchen | GridWorld | Crafter |
|---|---|---|---|---|---|
| Goal-conditioned BC | -2.63 | -27.06 | 0.09 | 0.05 | 11.78 |
| BC-finetune | -3.09 ± 0.12 | -43.19 ± 2.97 | 0.00 ± 0.00 | 0.04 ± 0.00 | 1.88 ± 0.19 |
| SPiRL | -2.62 ± 0.35 | -28.94 ± 0.79 | 0.22 ± 0.05 | 0.10 ± 0.03 | 10.96 ± 0.25 |
| PTGM | -0.96 ± 0.08 | -26.74 ± 1.71 | 0.25 ± 0.03 | 0.25 ± 0.03 | **15.72** ± 0.12 |
| MGPO | **-0.41** ± 0.10 | **-14.21** ± 0.19 | **0.63** ± 0.02 | **0.58** ± 0.03 | 15.66 ± 0.14 |

## 4.2 Baselines and Main Results

We compare MGPO with previous methods that focus on task-agnostic pre-training. Further details are provided in Appendix F.

**Goal-conditioned BC** feeds the task goal $g^M$ directly into the pre-trained goal-conditioned policy for new tasks. Its performance is akin to the initial prompt performance in MGPO without prompt optimization.

**BC-finetune** updates the parameters of the pre-trained model with RL based on online collected trajectories, which is a common method in previous work [4]. Conditioned on the task goal, we finetune the Transformer parameters using PPO [46].

**SPiRL** [38] pre-trains short-term skills along with their latent representations and uses online RL to train a high-level policy for online adaptation. **PTGM** [60] improves this approach by pre-training a goal-conditioned policy and discretizing the goal space for the high-level policy.

For each test task, we evaluated each method for 100 episodes of online rollout, measuring performance by the average return of the optimized policy or prompt. Results in Table 1 summarize the performance of MGPO against all baselines across different environments, where MGPO is implemented with UCB in online adaptation.

- **MGPO's superiority:** MGPO demonstrates superior performance in all environments, showcasing its efficiency in adapting to new tasks.

- **Limitations of BC-finetune:** This method underperforms others in most environments due to the instability and inefficiency of finetuning parameters in the entire model with online RL. It fails to solve tasks within the small budget of 100 online episodes.

- **Comparing SPiRL, PTGM, and MGPO:** While SPiRL and PTGM exhibit better sample efficiency than BC-finetune, they largely underperform MGPO in MazeRunner, Kitchen, and GridWorld. These approaches are limited by their reliance on learning a high-level RL policy for goal switching at different states during online adaptation. For example, PTGM trains a high-level policy $\pi(g|h_t, o_t)$ to select goals, which is effectively searching policies in the joint space of $O^K \times G$. In contrast, MGPO, with its inherent ability to seamlessly switch between goals, optimizes in the prompt space $G^K$, leading to more efficient adaptation.

- **Goal-conditioned BC:** The performance of this method highlights MGPO's capability to refine and improve upon the initial prompt. Conditioned on a single goal, the performance of Goal-conditioned BC is sub-optimal due to limitations in dataset quality (e.g., MazeRunner) and the partial observability in task specifications (Kitchen and GridWorld) and environment layouts (MazeRunner and Crafter). In contrast, through online exploration, MGPO gathers more task and environment information in the unseen task and can search for a prompt sequence to stitch different short-term behaviors learned from data.

- **Performance in Crafter:** Performance of Goal-conditioned BC, PTGM, and MGPO in Crafter is similar compared with other environments. This is because, the dataset does not feature great diversity since the data collection policy always aims at unlocking more achievements, thereby the pre-trained policy may be less sensitive to the prompt. In contrast, in other environments, the prompt substantially influences the policy's behavior and performance.

**Table 2:** Performance of MGPO with different prompt optimization methods. Each result shows the average performance on all test tasks in the environment and the standard deviation across 3 random seeds for online test.

| Method | MazeRunner-15 | MazeRunner-30 | Kitchen | GridWorld | Crafter |
|---|---|---|---|---|---|
| MGPO-GRIPS | $-0.56 \pm 0.09$ | $-13.01 \pm 1.22$ | $0.54 \pm 0.09$ | $0.40 \pm 0.01$ | $15.53 \pm 0.02$ |
| MGPO-BBT | $-0.80 \pm 0.16$ | $\mathbf{-12.72} \pm 2.22$ | $\mathbf{0.74} \pm 0.06$ | $0.42 \pm 0.03$ | $\mathbf{16.31} \pm 0.03$ |
| MGPO-explore | $-0.71 \pm 0.03$ | $-14.74 \pm 0.49$ | $0.47 \pm 0.05$ | $0.48 \pm 0.01$ | $15.82 \pm 0.07$ |
| MGPO-UCB | $-0.41 \pm 0.10$ | $-14.21 \pm 0.19$ | $0.63 \pm 0.02$ | $\mathbf{0.58} \pm 0.03$ | $15.66 \pm 0.14$ |
| MGPO-BPE | $\mathbf{-0.38} \pm 0.05$ | $-14.86 \pm 1.32$ | $0.65 \pm 0.02$ | $0.57 \pm 0.01$ | $15.65 \pm 0.06$ |

**Table 3:** Performance of prompts optimized with different methods in Kitchen with noisy observations or actions. Each result shows the average performance on all test tasks and the decrease compared with the environment without noise.

| Method | MGPO-GRIPS | MGPO-BBT | MGPO-UCB | MGPO-BPE |
|---|---|---|---|---|
| Noisy observations | 0.29 (-0.25) | 0.42 (-0.32) | **0.49** (**-0.14**) | 0.45 (-0.20) |
| Noisy actions | 0.12 (-0.42) | 0.19 (-0.55) | **0.30** (**-0.33**) | 0.28 (-0.37) |

## 4.3 Prompt Optimization Methods

In exploring prompt optimization methods for the pre-trained multi-goal Transformer, we draw insights from the area of prompt optimization for language models, including two contemporary methods to implement MGPO:

**GRIPS** [40] is a genetic algorithm designed for prompt search within the discrete prompt space. In our implementations, we start with an initial prompt and generate new prompts to evolve it via online evaluating all prompts. We implement operations of adding, deleting, and swapping goals for generating new prompts. The algorithm returns the best-evaluated prompt.

**BBT** [48] is a black-box optimization method for continuous prompt-tuning. It utilizes a low-dimensional vector $z$, which is added to the initial prompt through random projection. This vector is then optimized using a CMA-ES [19] evolution strategy. Since it extends the prompt into a continuous space, the optimized prompts may not be interpreted as sequences of discrete goals.

For our proposed method, we investigate the two options for the MAB algorithm: **UCB** and **BPE**. Additionally, we examined an ablation approach dubbed **MGPO-explore**, which differs from our MAB formulation by leveraging the most recent prompt sampled from the highest-return trajectory for further exploration.

Table 2 presents experimental results of MGPO implemented with different prompt optimization methods. Across all environments, all MGPO methods demonstrate superior performance compared to the baseline methods in Table 1, showing MGPO's great compatibility with different prompt optimization methods.

UCB and BPE consistently outperform MGPO-explore in most environments. This result demonstrates the efficacy of our MAB formulation in the prompt optimization process. By exploring and exploiting existing prompts, MAB-based methods yield trajectories with higher returns, thus improving the overall performance.

UCB and BPE outperform the discrete search method GRIPS in four out of the five environments. This result highlights the strength of our proposed methods in discrete optimization for multi-goal prompts. While UCB and BPE outperform BBT in MazeRunner-15 and GridWorld, a reversal occurred in the other three environments, particularly in Kitchen where BBT surpasses the performance of all other methods. We speculate that, due to the nature of deterministic and differentiable transitions in Kitchen, the continuous optimization method BBT may quickly find a local optimum.

The performance of BBT raises critical considerations about the nature of the optimized prompts. Despite its efficiency, the optimized continuous prompts deviate significantly from the training distribution, potentially harming the robustness of the policy. We conduct further studies to evaluate the robustness of the optimized prompts. Table 3 compares the performance of the optimized prompts

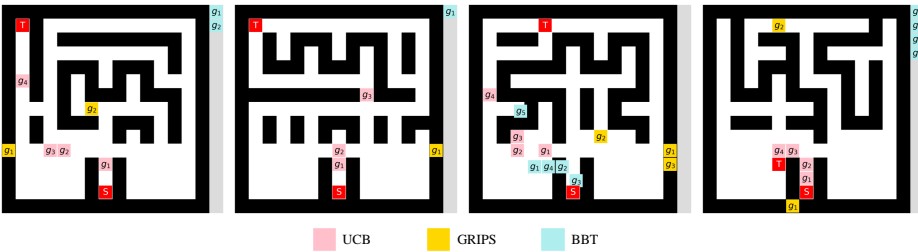

**Figure 2:** Visualization of the optimized prompts in four MazeRunner-15 tasks. The prompt with each method is displayed in a unique color. $S$ and $T$ represent the start position and the task goal respectively, and $g_1, g_2, ...$ represents goal positions in the prompt. We display the goals that exceed the maze boundaries in the gray bar on the right side.

**Table 4:** Results of ablation study on MGPO with varying maximal prompt length $K$ and hyperparameter $c$ in UCB.

| Ablation | MazeRunner-15 | MazeRunner-30 | Kitchen |
|----------|---------------|---------------|---------|
| $K = 1$ | -2.63±0.00 | -27.06±0.00 | 0.09±0.00 |
| $K = 2$ | -1.13±0.10 | -19.57±1.21 | 0.28±0.00 |
| $K = 3$ | -0.77±0.04 | -17.12±0.47 | 0.48±0.02 |
| $K = 5$ | -0.41±0.10 | -14.21±0.19 | 0.63±0.02 |
| UCB-0 | -0.44±0.07 | -15.86±0.51 | 0.63±0.05 |
| UCB-1 | -0.41±0.10 | -14.21±0.19 | 0.63±0.02 |
| UCB-10 | -0.40±0.04 | -14.70±0.66 | 0.66±0.02 |

when the robotic environment Kitchen has noisy observations or actions. It reveals that prompts with BBT and GRIPS are particularly susceptible to environment perturbation, hinting at out-of-distribution prompts. In contrast, prompts with our proposed methods maintain robust performance facing perturbation.

Figure 2 visualizes goals in prompts optimized by different methods in MazeRunner-15. It clearly showcases the interpretability of the prompt with our proposed methods, which represents meaningful waypoints toward the task goal. In contrast, prompts with GRIPS and BBT lack this level of clarity and interpretability.

### 4.4 Ablation Study

In addition to MGPO-explore, we conduct more ablation studies to examine the impact of components in MGPO. The results of ablation studies are presented in Table 4.

**Maximal Prompt Lengths:** We study the impact of different maximal prompt lengths $K$ during online adaptation. Complex tasks often necessitate multiple goals within a prompt to specify varying behaviors in long-horizon trajectories. We observe that optimizing prompts with increasing length consistently improves performance. Specifically, a length of $K = 5$ yields the best results, and reducing $K$ to 1 results in performance akin to that of Goal-conditioned BC. Appendix B.2 shows additional results when further increasing $K$.

**UCB Hyperparameters:** UCB includes a hyperparameter $c$ that balances exploration and exploitation (Appendix E.1). In our test with MGPO-UCB under varying values of $c$, we observe that the performance exhibits low sensitivity to this hyperparameter's selection, as shown in Table 4.

**Dataset Quality:** We also assess the influence of dataset quality on MGPO's performance. Detailed results are provided in Appendix B.3.

### 4.5 Visualization and Case Study

In MazeRunner, we visualize the evolution of MGPO during online adaptation. Figure 3 displays the progression of both the prompt and policy behavior at various stages. At the beginning, the policy is

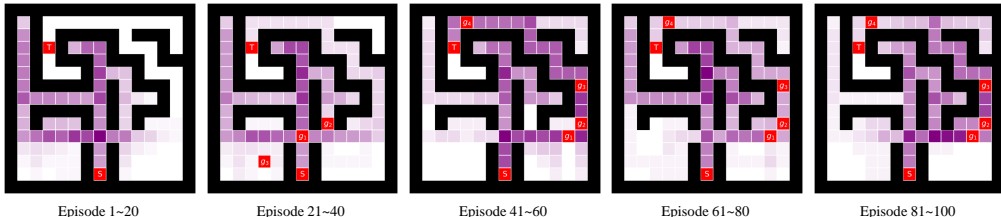

| Episode 1~20 | Episode 21~40 | Episode 41~60 | Episode 61~80 | Episode 81~100 |

**Figure 3:** Visualization of the optimized prompts and state visitation in the adaptation stage of MGPO-UCB. Red squares with $S, T, g_i$ represent the start positions, task goals, and the optimized prompts, respectively. We use purple to display the visitation frequency of each location, with darker shades indicating higher frequencies. In this example, initially, the agent explores the left half of the maze, aiming for the front-left task goal but is hindered by walls. In exploration, as it discovers rewarding routes to the right, MGPO-UCB adapts by sampling new prompts from these better paths. An optimized prompt is achieved after 40 episodes.

conditioned on a short prompt, resulting in high action uncertainty and exploratory behavior, where the policy explores for diverse trajectories toward the task goal. This exploratory phase is crucial, as it discovers trajectories with higher returns toward the goal and helps in understanding the unknown task environment. As the algorithm progresses, the prompts are iteratively refined based on returns from the environment, and the policy behavior is steered towards more efficient paths. More results are presented in Appendix B.4.

## 5 Related Work

**Policy Pre-Training for RL** is a topic studying learning from datasets to enhance the efficiency of RL on new tasks. Offline meta-RL (OMRL) focuses on learning to adapt to new tasks with a few samples, employing context-based learning methods [26, 59, 39] or gradient-based meta-learning [33]. However, OMRL necessitates multi-task datasets for training, requiring extensive trajectory collection within each task. On the other hand, task-agnostic pre-training leverages rich behaviors in task-agnostic datasets, exploring imitation learning [43, 4], offline RL [24, 54], or hierarchical skill learning [38, 45, 60] for policy pre-training. Our work addresses the challenging setting of task-agnostic pre-training for long-horizon tasks. Unlike previous methods that rely on RL for online adaptation, our proposed multi-goal Transformer enables RL-free optimization methods, enhancing sample efficiency.

**Transformers for RL.** The Transformer architecture has become increasingly popular in RL for its ability in sequence modeling and long-term memory. In offline RL, Decision Transformers [8, 22, 14, 57] recast RL as a sequence modeling problem conditioned on return-to-go. In context-based meta-RL, Transformers are employed to handle multiple trajectories, modeling task adaptation as in-context learning [23, 56, 29]. In multi-task RL, large-scale Transformers are adopted with extensive datasets to address complex robotic domains [6, 5] and generalization to various tasks [44, 49, 15]. In our work, we utilize Transformers to address the demands of long-term memory in modeling long-horizon trajectories with multiple goals.

**Prompt Optimization** for pre-trained Transformer-based language models has demonstrated its effectiveness in adapting to downstream NLP tasks without tuning model parameters. It optimizes the prompt, which is a sequence of tokens input to the model specifying the task. In the recent large language models (LLMs) [7], task adaptation can be easily achieved by prompt design, with the techniques of in-context learning [11] and chain-of-thought reasoning [52]. For earlier language models, prompt-tuning methods optimize prompts in a continuous space [25, 27], while prompt search methods focus on optimizing discrete prompt tokens for interpretability [9, 40]. We adopt the concept of prompt optimization to address online adaptation in RL, focusing on optimizing the goal sequence for the pre-trained Transformer. Unlike in NLP tasks, our approach places a unique emphasis on sample efficiency, a crucial aspect given the necessity of online evaluation.

# 6 Conclusion and Limitations

We propose MGPO, a novel framework for policy pre-training to enhance online adaptation in unseen long-horizon tasks. By integrating the strengths of Transformer architectures and goal-conditioned policies during pre-training, MGPO enables an efficient prompt optimization process in the online adaptation phase. Our extensive experimental results across various environments show MGPO's superiority over existing methods. A comparative analysis of different prompt optimization techniques highlights the advantages on interpretability and robustness of our method over other contemporary approaches.

The effectiveness of MGPO in solving long-horizon tasks opens new possibilities for real-world applications where efficient online adaptation is crucial. However, our experiments have been confined to simulated environments so far. Future research can focus on scaling MGPO to larger pre-training datasets and testing it in more complex, real-world environments.

Like many offline RL approaches, the performance of MGPO is influenced by the quality of the dataset used during pre-training. To address this limitation, future work could explore several potential directions: incorporating online data collection to improve dataset quality, using offline RL methods such as DT to train the multi-goal Transformer, and integrating MGPO with finetuning methods to enhance its adaptability.

The prompt-based policy in MGPO, similar to language models, can exhibit unpredictable behavior when encountering out-of-distribution prompts. Furthermore, even minor alterations to the prompt may lead to unintended behaviors, raising safety and robustness challenges. Future efforts could explore enhancing the robustness of prompt-based policies.

### Acknowledgments

This work was supported by NSFC under Grant 62450001 and 62476008. The authors would like to thank the anonymous reviewers for their valuable comments and advice.

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

# A    Illustrations of MGPO

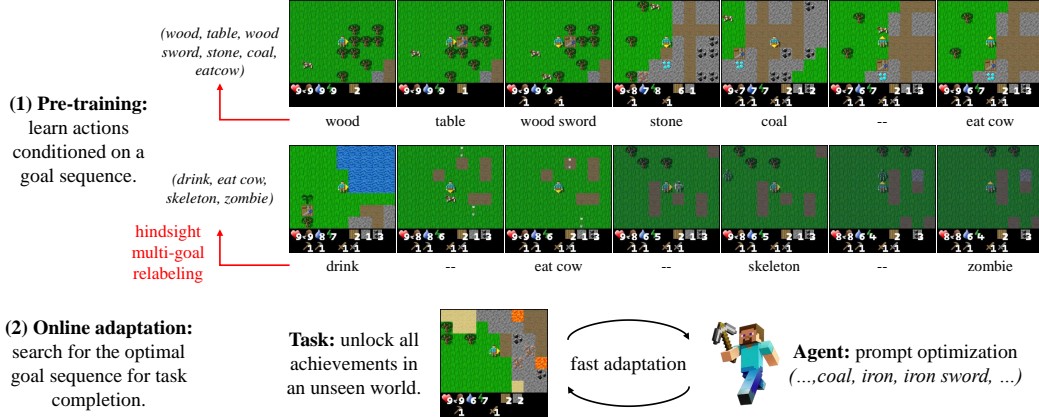

*(wood, table, wood sword, stone, coal, eatcow)*

**(1) Pre-training:** learn actions conditioned on a goal sequence.

*(drink, eat cow, skeleton, zombie)*

hindsight multi-goal relabeling

**(2) Online adaptation:** search for the optimal goal sequence for task completion.

**Task:** unlock all achievements in an unseen world.

fast adaptation

**Agent:** prompt optimization *(...,coal, iron, iron sword, ...)*

**Figure 4:** A running example illustrating MGPO. During the pre-training stage, the agent learns to complete arbitrary sequences of goals using offline trajectories with hindsight-relabeled goals. In the online adaptation stage, the agent optimizes the goal sequence to maximize returns in an unseen environment.

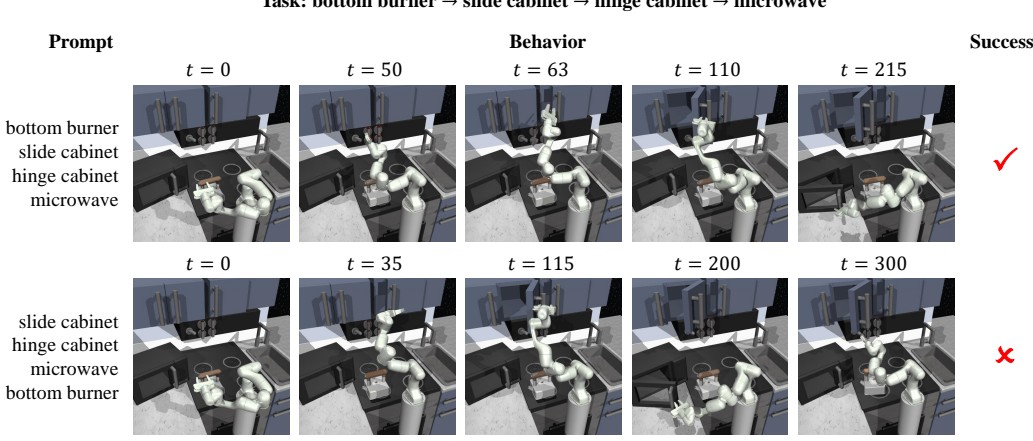

**Figure 5:** An illustration of prompts (goal sequences) and the agent's behavior in a Kitchen task. The order of goals within the prompt is crucial for successful task completion.

# B    Additional Experimental Results

## B.1    Learning Curves During Online Adaptation

Figure 6 shows the learning curves of MGPO and baselines during online adaptation. Figure 7 shows the learning curves of MGPO with different prompt optimization methods during online adaptation.

## B.2    Ablation Study on the Maximal Prompt Length

Since we use $K = 5$ in the pre-training phase and the model has not encountered prompts exceeding a length of 5, we present results with maximal prompt lengths $K \leq 5$ in our main paper. Here, we present additional results when further increasing $K$. We pre-train models with $K = 10, 20$ and $40$ to test online adaptation with larger prompt lengths.

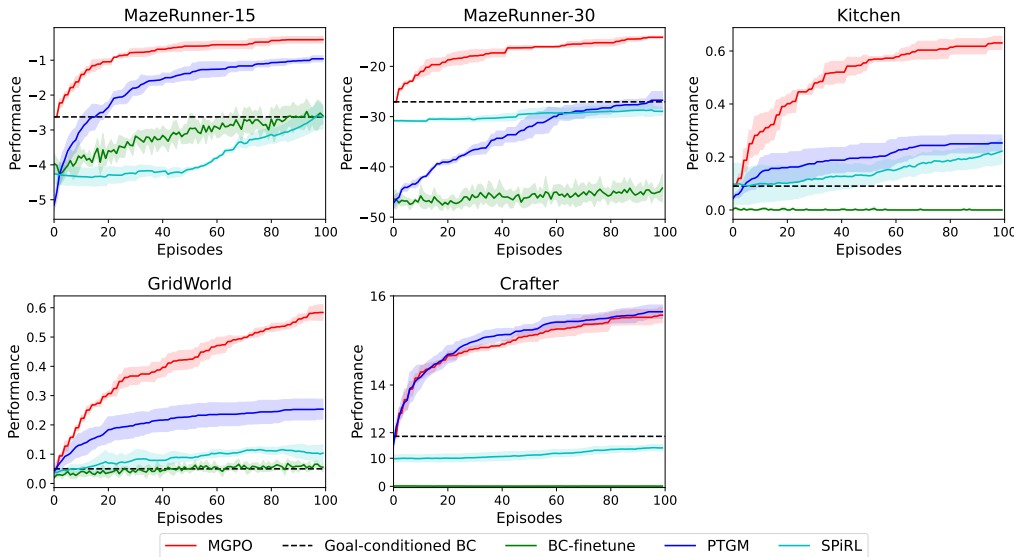

**Figure 6:** Performance of MGPO compared with baseline methods during online adaptation. The vertical axis indicates the task performance of the optimized policy and the horizontal axis indicates the number of online episodes. Each figure shows the average performance on all test tasks in each environment and the standard deviation across 3 random seeds.

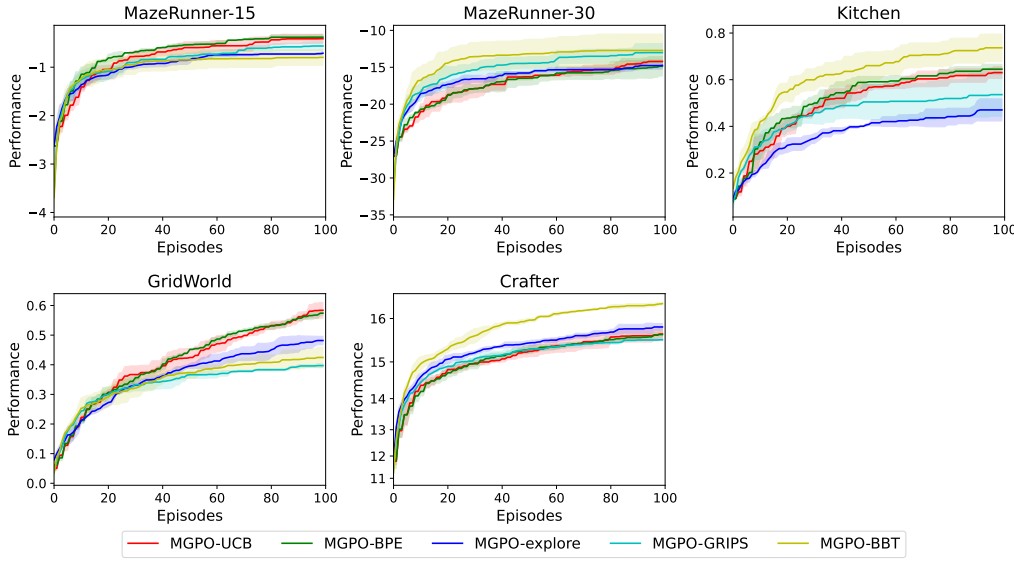

**Figure 7:** Performance of MGPO with different prompt optimization methods during online adaptation.

Figure 8 shows the results. In MazeRunner-15, setting $K = 10$ yields the optimal performance. Increase beyond this value leads to a decline in learning efficiency. In MazeRunner-30, performance exhibits a slight improvement as $K$ extends to 40. This suggests that for more extended, long-horizon tasks, the exploration of longer prompts can be advantageous.

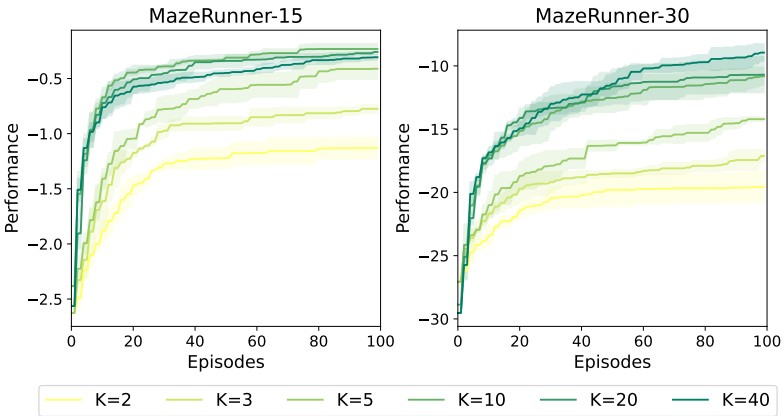

**Figure 8:** Additional results of ablation study on the maximal prompt length $K$.

### B.3    Ablation Study on Dataset Quality

We assess MGPO's performance with datasets of varying quality in MazeRunner. In Table 5, we denote datasets collected using an A* algorithm with a maximum of $n$ goal switches per episode as A*-$n$, and datasets from a random exploration policy as Random. Results show that MGPO achieves better performance trained on A*-2 datasets than A*-1, indicating its efficacy with data containing diverse long-horizon behaviors. The comparatively lower performance on the A*-4 dataset in MazeRunner-30 and Random datasets suggests MGPO's reliance on the quality of data collection policies. Future work could involve integrating offline RL methods to boost performance with lower-quality datasets.

**Table 5:** Results of ablation study on varying dataset quality, evaluated with MGPO-UCB.

| Dataset | MazeRunner-15 | MazeRunner-30 |
|---------|---------------|---------------|
| Random  | -1.05±0.01    | -17.13±0.52   |
| A*-1    | -0.97±0.03    | -9.26±0.38    |
| A*-2    | -0.36±0.02    | -8.66±0.35    |
| A*-4    | -0.41±0.10    | -14.21±0.19   |

### B.4    Additional Visualization Results

By prompting the pre-trained policy with prompts of varying lengths, MGPO can naturally trade-off exploration and exploitation during online adaptation. To empirically demonstrate this capability, we visualize prompts of different lengths and the corresponding behaviors of the pre-trained policy in Figure 9. We observe that a shorter prompt, by providing less information to the policy, introduces higher uncertainty and thus encourages more exploratory behaviors. Conversely, a longer prompt, containing more detailed information from the trajectory, prompts more deterministic and exploitatory behavior.

Figure 10 shows additional results on the progression of both the prompt and policy behavior at various stages.

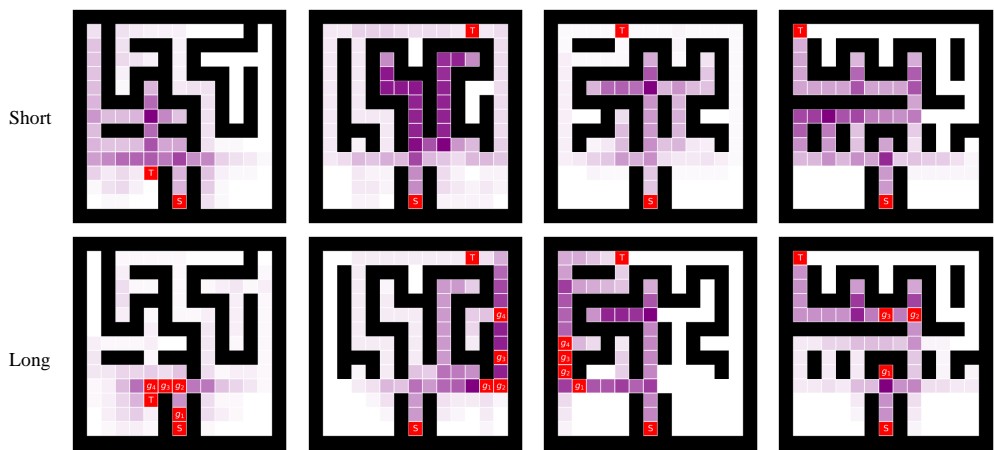

**Figure 9:** Visualization of prompts with different lengths and their state visitation counts in MazeRunner-15. Each image in the upper row shows a short prompt with a length of 1. Each image in the bottom row shows a long prompt with a length of 5.

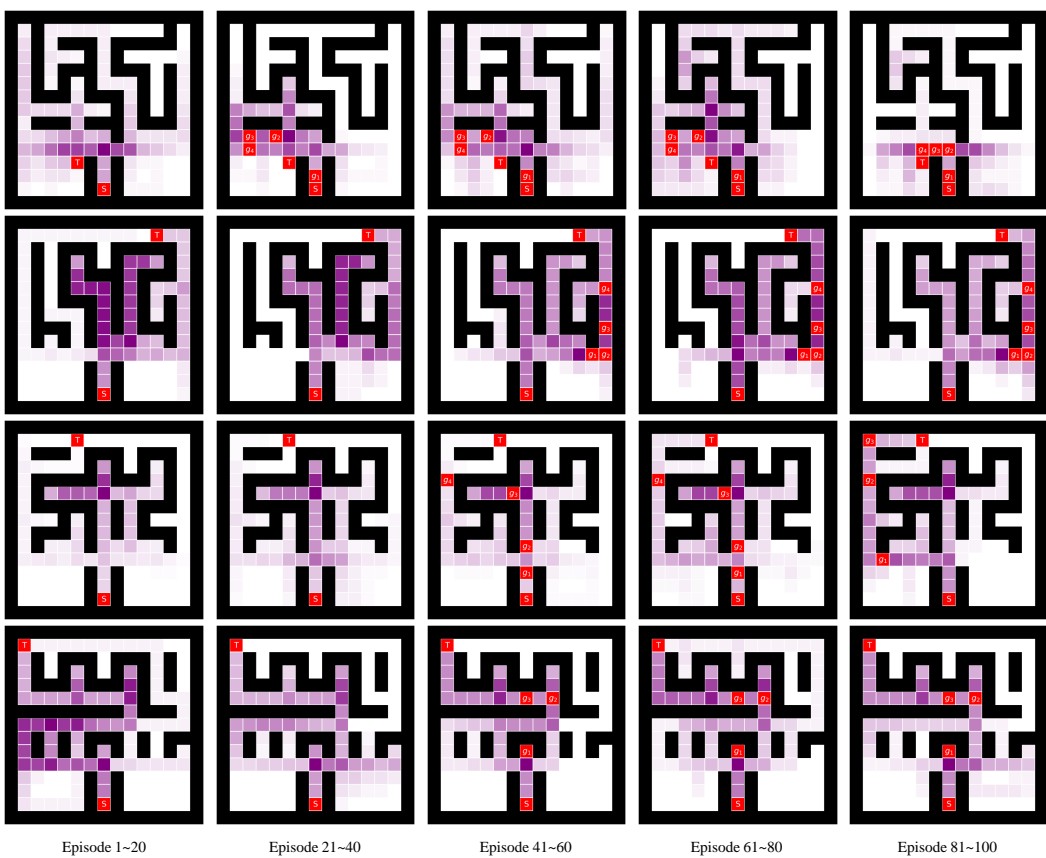

| Episode 1~20 | Episode 21~40 | Episode 41~60 | Episode 61~80 | Episode 81~100 |
|---|---|---|---|---|

**Figure 10:** Additional results on visualization of the optimized prompts and state visitation in the adaptation stage of MGPO-UCB. Each row represents a task in MazeRunner-15.

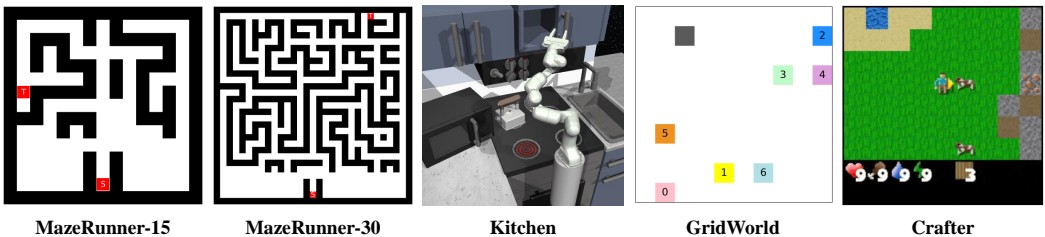

| MazeRunner-15 | MazeRunner-30 | Kitchen | GridWorld | Crafter |

**Figure 11:** Environments in our experiments.

## C   Details for Environments and Data Collection

### C.1   MazeRunner

MazeRunner is a 2D maze navigation environment implemented in AMAGO [15] and is mainly based on Memory Maze [37]. The agent navigates in a grid world of size $N \times N$, where we use $N = 15$ and $30$ in our experiments. For each task, the unknown structure of walls in the maze is distinct, and the agent should move to the provided goal location $g^M = (x^M, y^M)$ to accomplish the task. The agent receives 6-dimensional observations $(x, y, d_1, d_2, d_3, d_4)$, where the first two values are the agent's position and the last four values are its distances to walls in the four directions. We use the agent's positions in observations to represent goals. The discrete action space consists of moving left, right, forward, backward, and no movement. The reward is $+1$ when the agent reaches the task goal and a penalty of $-0.1$ is given at each step in addition. The task horizon is 64 steps for MazeRunner-15 and 500 steps for MazeRunner-30. The episode terminates when the agent reaches the task goal or reaches the task horizon.

For data collection, we implement a rule-based policy that repeatedly samples a goal position in maze and plan for a path to reach it with A* algorithm. The A* algorithm takes the partial observations of the agent, estimate a score for each position in the known part of the maze, and returns a shortest path towards the position with highest score. In each episode, the maze is randomly generated and the policy samples a new goal when achieving the last goal for 4 times, until achieving all goals or the environment terminates. We collect 10K trajectories for both MazeRunner-15 and MazeRunner-30. We sample 50 tasks of different mazes and task goals for test.

### C.2   Kitchen

The Kitchen [16] environment is designed to simulate a kitchen scenario for indoor robotics with several interactive objects. The subtasks used in this paper include activating the bottom bunner, switching on the light, opening the microwave, sliding the slide cabinet, and opening the hinge cabinet. The target of each subtask is to adjust the corresponding object to a target pose. The observation is a 59-dimensional vector, representing positions and velocities of all objects and the 7-DoF robot arm. The actions are 9-dimensional continuous vectors representing velocities to control the motor actuators in the robot arm. Each task is defined with a set of $n$ subtasks among the five subtasks and the order to complete them. The task horizon is set to 500 timesteps. When a subtask is completed in the correct order, the environment provides a reward of $+\frac{1}{n}$. The goal space consists of 5-dimensional binary vectors representing each subtask in each bit. For the task goal revealed to the agent, bits of the selected subtasks are set to 1 and other bits are set to 0. Other goals are one-hot vectors representing the current subtask to complete.

We verified that the Kitchen datasets provided in D4RL [13] do not contain diverse transitions between the five subtasks. To collect trajectories completing different sets of subtasks, we use PPO [46] to train a policy for each sub-task using a shaped reward function and varied initial states from the Kitchen-mixed-v0 dataset [13]. We sample 300 tasks for data collection. In each task, we chain the policies learned with PPO to sequentially solve subtasks and collect 30 episodes. The whole dataset consists of 9K trajectories. We sample a test set of 25 tasks, which has no overlap to tasks used in data collection.

### C.3 GridWorld

This environment is a $10 \times 10$ grid world with 7 distinct objects at different locations. Each object has a binary state, which defines a subtask that is completed when the state turned to 1 from the initial state of 0. The agent observes a 9-dimensional vector representing its location and states of objects. The discrete action space consists 6 actions: moving to the four directions, flip the object state, and no operation. Since this environment is a simplified version of Kitchen, tasks and goals are defined similar to Kitchen's. The task horizon is 90 in this environment.

We use a rule-based expert policy to collect 13K trajectories and sample a test set of 50 unseen tasks.

### C.4 Crafter

Crafter [18] is a research-friendly benchmark mirroring the popular open-world survival game Minecraft. The 2D world map with infinitely large size is procedurally generated. The observations are $64 \times 3$ rgb images showing the first-person view of the agent, which contains a $9 \times 9$ part of the world and the rendered information of inventory and life. The discrete space contains 17 actions, including movement, attack, tool use, and crafting. There are 22 achievements in the game. When the agent unlocks each achievement for the first time, it receives a reward of $+1$. When the agent's health increases or decreases by one point, it receives a corresponding reward of $+0.1$ or $-0.1$. The target of the task is to complete all achievements. Different tasks are featured with different generated world maps, which can be specified with a random seed. Goals are 22 dimensional binary vectors representing each achievement with each bit. For hindsight relabeling, the task goal is set to the accomplished achievements in each trajectory. For online test, we set the task goal to all-one to encourage more achievements. Other goals are one-hot vectors indicating the current achievement to unlock.

We set the task horizon to 500 steps. We use the policy trained in Achievement Distillation [35] to collect a dataset of 2K trajectories. We use 50 test task specified with random seeds unused in data collection. The average episode return in the dataset is 12.3 and our method can increase this to 16 in online adaptation.

## D  Training Details

We build our Transformer architecture based on Decision Transformers [8], using a lightweight backbone of GPT-2 [42] with 0.6M parameters. Input tokens of $o$, $a$, and $g$ are transformed into 128-dimensional embedding vectors using three different linear layers, respectively. We use a positional embedding that embeds the timestep of each token and adds it to the token embedding, enhancing the timestep information of inputs. The Transformer then processes the sequence of embeddings and outputs a sequence with the same dimension and length. We use a linear layer to decode tokens at positions of the input tokens of observations to predict actions and compute loss. To train on batches of trajectories, we pad the input prompts and trajectories to lengths of $K$ and $H$, respectively.

In MazeRunner, we sample prompts from the agent's locations in the whole trajectory. To augment the diversity of task goals and trajectory lengths, we truncate the trajectory at a random timestep $h$ for each sampled trajectory and use $o_h$ to represent its task goal. In other environments, goals in prompts are sampled from the completed subtasks in each trajectory and $g_H$ represents the actual set of subtasks completed at the end of each trajectory.

Table 6 lists the hyperparameters used in pre-training.

All models are trained on a lab machine with a single NVIDIA RTX 4090 GPU and Intel i9 CPUs. For each environment, the pre-training stage takes about 12 hours.

## E  Details for Prompt Optimization

### E.1  MGPO-UCB and MGPO-BPE

We summarize our proposed prompt optimization algorithm in Algorithm 1.

**Table 6:** Hyperparameters used in pre-training for all environments.

| Name | Value |
|---|---|
| Embedding dimension | 128 |
| Number of layers | 3 |
| Number of attention heads | 1 |
| Activation | ReLU |
| Batch size | 64 |
| Learning rate | 1e-4 |
| Learning rate decay weight | 1e-4 |
| Dropout | 0.1 |
| Warmup steps | 10000 |

---

**Algorithm 1** Prompt Optimization in MGPO-UCB and MGPO-BPE

---

**Input:** pre-trained policy $\pi_\theta$; task goal $g^M$; interaction budget $N$
**Initialize:** prompt buffer $B = \{(g^M) : \emptyset\}$; best return $R^* = -\infty$ and best trajectory $\tau^* = \emptyset$
**while** not exceeding $N$ episodes **do**
   Select $p^*$ from $B$ with MAB algorithms.
   Collect $\tau$ and get return $R$ with $\pi_\theta(a|p^*, h, o)$.
   Update $B$, $R^*$, $\tau^*$.
   Sample new prompt $p' \sim \tau^*$.
   Replace the last goal in $p'$ with $g^M$.
   Collect $\tau$ and get return $R$ with $\pi_\theta(a|p', h, o)$.
   Update $R^*$, $\tau^*$.
   **if** $R > \max_{p \in B} \bar{R}_p$ **then**
      Insert $(p' : \{R\})$ to $B$.
   **end if**
**end while**

---

Here we present the MAB algorithms used in our methods for prompt selection. At each iteration, we have a buffer of prompts $B = p$, where each prompt $p$ maintains its historical returns $(R_{p,1}, \cdots, R_{p,n_p})$. We aim to select a $p^*$ to collect trajectories of high returns.

**UCB:** We estimate the upper confidence bound of the expected return for each prompt:

$$\text{UCB}(p) = \bar{R}_p + c\sqrt{\frac{\log(N)}{2n_p}}, \tag{3}$$

where $\bar{R}_p$ is the mean return in history, $c$ is a hyperparameter which is set to 1 in our main results, and $N = \sum_{p \in B} n_p$. The prompt for online exploration is selected with $p^* = \text{argmax}_{p \in B} \text{UCB}(p)$.

**BPE:** We assume a Gaussian prior distribution $\mathcal{N}(\mu_0, \sigma_0)$ for returns of each prompt. We set $\mu_0 = \frac{1}{2}(R_{\min} + R_{\max})$ and $\sigma_0 = \frac{1}{2}(R_{\max} - R_{\min})$, where $R_{\min}$ and $R_{\max}$ are the minimal and maximal possible returns in the environment, respectively. Given historical returns of each prompt, we estimate its posterior distribution $\hat{P}_p(R) = \mathcal{N}(R|\mu_p, \sigma_p)$ with a Bayesian approach:

$$\frac{1}{\sigma_p^2} = \frac{1}{\sigma_0^2} + \frac{n_p}{\sigma_{R_p}^2}, \quad \mu_p = \sigma_p^2 \left( \frac{\mu_0}{\sigma_0^2} + \frac{n_p \bar{R}_p}{\sigma_{R_p}^2} \right), \tag{4}$$

where $\bar{R}_p$ is the mean historical return and $\sigma_{R_p}$ is the standard deviation of historical returns. We select the prompt which has the maximum probability to yield returns exceeding the current highest return: $p^* = \text{argmax}_{p \in B} \hat{P}_p(R > R^*)$.

## E.2 GRIPS

GRIPS [40] is a gradient-free, search-based algorithm for prompt optimization. At each iteration, it performs editing operations on the prompt and evaluates the edited prompts. We implement three operations to edit the prompt. **Add:** Randomly sample a goal and insert it into the current prompt at a random position. **Del:** Randomly delete a goal in the current prompt. **Swap:** Randomly select two goals and swap them in the current prompt.

We initialize the prompt with $p_0 = (g^M)$ and maintain $g^M$ at the last position of the prompt while editing the preceding goals. In each iteration, we randomly generate $n = 5$ edited prompts, online evaluate their performance, and preserve the prompt with the highest return.

We summarize GRIPS in Algorithm 2.

---

**Algorithm 2** GRIPS

---

**Input:** pre-trained policy $\pi_\theta$; task goal $g^M$; interaction budget $N$; the number of edited prompts per iteration $n$
**Initialize:** best return $R^* = -\infty$ and best prompt $p^* = (g^M)$
**while** not exceeding $N$ episodes **do**
    Edit $p^*$ with random operations from {Add, Del, Swap} to generate prompts $\{p_i\}_{i=1}^n$.
    Online evaluate each $p_i$ with $\pi_\theta(a|p_i, h, o)$, get return $R_i$.
    $i^* = \operatorname{argmax}_i R_i$.
    **if** $R_{i^*} > R^*$ **then**
        $R^* = R_{i^*}, p^* = p_{i^*}$.
    **end if**
**end while**

---

## E.3 BBT

BBT [48] is a black-box continuous optimization method for prompt-tuning. We use $p = (g^M)$ to collect a trajectory $\tau$ and sample a initial prompt with $p_0 \sim \tau$. In the following iterations, we optimize $z \in \mathbb{R}^d$ in a low-dimensional space ($d < D$, where $D$ is the dimension of prompt) and use a random projection matrix $\mathbf{A} \in \mathbb{R}^{D \times d}$ to project $z$ and add to the initial prompt. Our objective is:

$$z^* = \operatorname*{argmax}_{z \in \mathbb{R}^d} R_{\mathbf{A}z + p_0}, \tag{5}$$

where $R$ is the expected return of the policy with prompt $\mathbf{A}z + p_0$.

BBT uses CMA-ES for optimization. In each iteration $t$, we sample $\lambda = 10$ new vectors $z^{(t+1)}$ from the multivariate normal distribution $z_i^{(t+1)} \sim m(t) + \sigma(t)\mathcal{N}(0, \mathrm{C}(t)), i = 1, ..., \lambda$, where $m(t) \in \mathbb{R}^d$ is the mean vector of the search distribution at iteration $t$, $\sigma(t) \in \mathbb{R}_+$ is a standard deviation controlling the step length, and $\mathrm{C}(t) \in \mathbb{R}^{d \times d}$ is a covariance matrix that determines the shape of the distribution. $m(t), \sigma(t),$ and $\mathrm{C}(t)$ are updated to maximize return.

We summarize BBT in Algorithm 3.

---

**Algorithm 3** BBT

---

**Input:** pre-trained policy $\pi_\theta$; task goal $g^M$; interaction budget $N$; the population size $\lambda$
**Initialize:** best return $R^* = -\infty$; best prompt $p^*$; $m(0), \sigma(0), \mathrm{C}(0),$ and $\mathbf{A}$
Collect an episode $\tau$ with $p = (g^M)$.
Sample initial prompt $p_0 \sim \tau$.
**while** not exceeding $N$ episodes **do**
    Sample $z_i, i = 1, ...\lambda$ from the multivariate normal distribution.
    Collect $\tau_i$ and get return $R_i$ with $\pi_\theta(a|\mathbf{A}z_i + p_0, h, o)$ for each $i$.
    Update $R^*$ and $p^*$.
    Update $m, \sigma, \mathrm{C}$ with CMA-ES.
**end while**

---

**Table 7:** Hyperparameters in BC-finetune.

| Name | Value |
|------|-------|
| Discount factor $\gamma$ | 0.98 |
| Learning rate $\alpha$ | 1e-4 |
| Learning rate decay | 1e-4 |
| Batch size $n$ | 5 |

**Table 8:** Performance of the BC-finetune baseline with different RL algorithms.

| Method | MazeRunner-15 | MazeRunner-30 | Kitchen | GridWorld | Crafter |
|--------|---------------|---------------|---------|-----------|---------|
| REINFORCE | $-2.59 \pm 0.24$ | $-44.22 \pm 2.75$ | $0.00 \pm 0.00$ | $0.06 \pm 0.02$ | $0.48 \pm 1.09$ |
| PPO | $-3.09 \pm 0.12$ | $-43.19 \pm 2.97$ | $0.00 \pm 0.00$ | $0.04 \pm 0.00$ | $1.88 \pm 0.19$ |

## F  Details for Baselines

### F.1  BC-finetune

We fix the prompt $p = (g^M)$ and finetune the policy parameters using RL algorithms during online adaptation. We have experimented with the policy gradient method REINFORCE [53] and the modern actor-critic method PPO [46]. However, we find that both approaches significantly underperform compared to MGPO. Thus, we only report results using PPO in our main paper. Table 7 lists the hyperparameters used in our experiments. Table 8 presents the results for BC-finetune using both REINFORCE and PPO.

### F.2  SPiRL and PTGM

These methods pre-trains diverse short-term skills to accelerate online RL by providing temporal abstractions.

SPiRL [38] pre-trains a skill encoder $q_\mu(z|h_t, o_t, a_{t:t+k})$, a skill decoder $\pi_\theta(a_{t:t+k}|h_t, o_t, z)$, and a skill prior $p_\psi(z|h_t, o_t)$. $q_\mu$ and $\pi_\theta$ is trained using the framework of conditional variational autoencoders, while $p_\psi$ is optimized to match the posterior distribution of $q_\mu$. In this approach, the short term actions $a_{t:t+k}$ are encoded into 10-dimensional continuous latent skills $z$. For each downstream task, it trains a high-level RL policy $\pi_\phi^H(z|h, o)$ with SAC [17]. The policy is regularized with a KL divergence loss between the policy and the skill prior $p_\psi$. Table 9 lists the hyperparameters used in SPiRL.

**Table 9:** Hyperparameters in SPiRL.

| Hyperparameters | Value |
|-----------------|-------|
| Weight for the KL loss | 5e-4 |
| Low-level steps | 10 |
| Discount factor | 0.95 |
| Learning rate | 1e-3 |
| Batch size | 256 |
| Target network update interval | 10 |

PTGM [60] pre-trains a low-level, goal-conditioned policy $\pi_\theta(a|g, h_t, o_t)$ along with a goal prior $p_\psi(g|h_t, o_t)$. We slightly modify MGPO to implement PTGM. We remove the prompt before the trajectory and associate each observation with a goal which is obtained via hindsight relabeling. $\pi_\theta$ is optimized via goal-conditioned behavior cloning and $p_\psi$ is optimized to maximize the log-likelihood

of the relabeled goal. In online adaptation, we use the discrete goal space as the high-level action space. We use DQN [34] to train the high-level policy $\pi_\phi^H(g|h_t, o_t)$. Its architecture is an MLP with 3 layers and hidden layer dimensions of 128. We also incorporate a KL regularization reward between the policy and the goal prior $p_\psi$ to accelerate online RL. Table 10 lists the hyperparameters used in PTGM.

**Table 10:** Hyperparameters in PTGM.

| Hyperparameters | Value |
|---|---|
| Weight for the regularization reward | 0.5 |
| Low-level steps | 10 |
| Discount factor | 0.95 |
| $\epsilon$ for $\epsilon$-greedy exploration | 0.1 |
| Learning rate | 1e-3 |
| Batch size | 64 |
| Target network update interval | 10 |

## G   Comparison with Related Areas

**Meta-RL / Offline Meta-RL:** Meta-RL approaches [56, 21, 59] typically require access to multiple trajectories per task and are often limited to short-term tasks due to the challenges of few-shot task inference. In contrast, our approach focuses on pre-training from **task-agnostic, offline datasets** and adapting to **long-horizon tasks**.

**Skill-based RL:** Online skill discovery methods [12, 61, 31] primarily focus on learning short-term skills through online RL. In contrast, our work focuses on offline skill acquisition from diverse, task-agnostic datasets, as conceptualized in the literature [38, 47, 60]. These methods typically use hierarchical RL to learn a high-level policy during online adaptation. Our approach differs by avoiding the inefficiencies of online RL; instead, we utilize a pre-trained Transformer-based policy, enabling us to streamline the adaptation process through efficient prompt optimization.

## H   Broader Impacts

This research contributes to the field of pre-training for RL, with the goal of improving the efficiency of RL agents to learn new tasks. While MGPO offers promising advancements, it also carries inherent risks such as reward hacking, where agents exploit loopholes in the reward function to achieve unintended high rewards. This can lead to unsafe or undesirable behaviors, especially in real-world applications. To mitigate these risks, it is vital to design robust reward functions and incorporate safety measures during training. Furthermore, the deployment of RL systems, particularly in sensitive areas like healthcare or autonomous driving, raises significant ethical considerations. Ensuring these systems are used responsibly is paramount to prevent negative societal impacts.

