# OpenReview forum: "Pre-Trained Multi-Goal Transformers with Prompt Optimization for Efficient Online Adaptation"
_NeurIPS.cc/2024/Conference — NeurIPS 2024 poster_

### Official Review · Reviewer_YjJH · 2024-07-13

**Soundness:** 3
**Presentation:** 2
**Contribution:** 2
**Rating:** 6
**Confidence:** 3

**Summary:**

The authors apply the concept of prompt optimization to reinforcement learning by pre-training a transformer-based policy on a tasks-agnostic (offline) dataset, injected with subgoals, to fine-tune a meta-policy optimizing the trajectory of goals to accomplish a given task.

**Strengths:**

The authors propose an exciting approach combining tasks-agnostic pre-training for reinforcement learning with the training paradigms of transformer architectures and large language models.

**Weaknesses:**

Clarity and overall writing could be improved. Concretely, the introduction seems overly abstract. A running example might help to illustrate the intended motivation.
Also, the overall setting remains not entirely clear. Especially the concept of goal relabeling in a task-agnostic setup, requiring a function to map observations to goals, should be explained earlier, given that Fig. 1 is already referenced in the introduction. Furthermore, the overall structure could be improved, by switching sections 2 and 3.

Also, the definition of M could be improved, e.g., including H, to define a finite horizon MDP.  Also, g seems to map each observation to one specific goal, causing that neither multiple goals could be accomplished concurrently nor an observation with no accomplished goal could occur. Extending the definition of the goal function should by adding its type signature could enhance comprehensibility. Furthermore, basing the goal on the observation seems to restrict its capabilities vastly.

Regarding the general setting, the proposed approach seems to require data that already accomplishes the intended task, vastly reducing the overall impact.  Also, I am missing comparisons to other tasks-agnostic pre-training approaches, like DIAYN (Eysenbach et al. 2018), and a precise delimitation of tasks and goals.

**Questions:**

How does the proposed approach solve the exploration challenge typically coupled with long-horizon RL tasks?

What problem is solved if the behavior collecting the data can already solve the intended task?

**Limitations:**

Limitations could have been discussed more extensively, especially regarding the volatility of a prompt-based architecture.

---

> ### Author Rebuttal · Authors · 2024-08-07
>
> ## Thanks for your review! Here, we respond to your comments and address the issues. We hope to hear back from you if you have further questions!
>
> **Q1:** Paper writing issues.
>
> **A1:** Thank you for your constructive feedback! We've added a running example in Figure 2 of our attached PDF to clarify our approach: using a gameplay dataset from Crafter, we train a goal-sequence-conditioned policy by extracting sequences of unlocked achievements from each trajectory. This policy adapts to unseen environments by optimizing the sequence of achievements to unlock.
>
> Regarding the concept of goal relabeling in a task-agnostic setup, we will provide an early explanation in the text, illustrated by how we define goals using game achievements extracted from observations. We also acknowledge that swapping the order of Sections 2 and 3 would improve clarity, and we plan to adjust our manuscript accordingly in the revision.
>
> We appreciate your suggestions on including the finite horizon $H$ in the definition of $M$. Given that $M$ represents a partially observable MDP, integrating $H$ will not change its properties, as the timestep $t$ can be regarded as a part of the hidden state. Regarding the goal function, we allow observations without any achieved goals, such as the Kitchen and Crafter environments. We will revise its definition by adding a type signature. Our current framework doesn't handle concurrent goals, which could be a potential area for future exploration.
>
>
> **Q2:** Can the behaviors in the dataset already solve the intended task?
>
> **A2:** The Goal-conditioned BC baseline fails to solve the test tasks in our experiments, indicating that the behavior policies used to collect the datasets cannot solve these tasks.
>
> In our problem formulation, we do not require data accomplishing the test tasks. Instead, we use **task-agnostic** datasets containing diverse, meaningful behaviors in the environment. Our pre-training scheme allows the policy to learn how to efficiently adapt to **unseen tasks with unknown dynamics and subtasks**. As in the case of the Crafter game, the dataset includes **general gameplay behaviors** without targeted objectives. During testing, our model efficiently optimizes goal sequences for a new world map, determining the most effective order to unlock achievements. We highlight the capability of our approach in efficiently solving new tasks without the need for task-specific pre-training. This aligns with the literature [1,2,3].
>
>
> **Q3:** Comparisons to task-agnostic pre-training methods, like DIAYN.
>
> **A3:** DIAYN, an unsupervised RL method, addresses a different problem from ours by discovering diverse skills online without task-specific rewards. Our approach, in contrast, leverages **offline task-agnostic datasets** to pre-train skills for efficient adaptation to long-horizon tasks. We acknowledge DIAYN's relevance in skill discovery but note that our focus is on enhancing task adaptation efficiency using offline pre-trained skills. Our paper discusses closely related works such as SPiRL[1], PTGM[2], and SKiMo[3], which share similarities with our approach but often struggle with inefficiencies in online adaptation -- a challenge we address through our prompt optimization method.
>
> **Q4:** A precise delimitation of tasks and goals.
>
> **A4:** In our framework, a 'task' is defined with a POMDP and typically involves long-horizon features, requiring sequential execution of many sub-processes. A 'goal' is a description of the agent's achieved state, as defined by our goal function.
>
> For example, as detailed in Section 3.1, we describe an environment with rooms of unknown layouts where the task is to navigate to a target location. The agent should find the most efficient path, which involves reaching several specific waypoints as necessary goals. Here, the 'task' is to efficiently reach the target location, while each 'goal' represents a waypoint the agent reaches.
>
>
> **Q5:** How does the proposed approach solve the exploration challenge in long-horizon RL tasks?
>
> **A5:**  Our approach addresses the exploration issue in long-horizon tasks with two main strategies:
> - **Temporal abstraction**: The pre-trained goal-conditioned policy leverages goal sequences to reduce exploration to high-level goal achievements, rather than detailed action sequences. This abstraction simplifies exploration, allowing the policy to execute extensive sequences towards each goal. Typically, fewer than 10 goals are required, although up to 500 environment steps are needed in each task in our environments.
> - **Prompt optimization:** Rather than training a high-level policy for goal-switching, our approach optimizes the sequence of goals directly. This avoids the necessity of learning goal selection based on observations during online adaptation, thus simplifying the task complexity and enhancing exploration efficiency.
>
>
> **Q6:** Limitations regarding the volatility of a prompt-based architecture.
>
> **A6:** We acknowledge the limitations of the prompt-based architecture. Just as with language models, our approach can perform unpredictably when faced with out-of-distribution prompts. Additionally, subtle changes to the prompt could lead to unintended behaviors, posing challenges in terms of safety and robustness. In our manuscript revision, we will further discuss these limitations.
>
> [1] Pertsch et al., Accelerating reinforcement learning with learned skill priors, 2021.
>
> [2] Yuan et al., Pre-training goal-based models for sample-efficient reinforcement learning, 2024.
>
> [3] Shi et al., Skill-based model-based reinforcement learning, 2023.

---

> > ### Comment · Reviewer_YjJH · 2024-08-12
> >
> > Thank you for your extensive response and for providing additional explanations and clarifications. My concerns are mostly addressed, and I will raise my score accordingly.

---

> > > ### Author Response · Authors · 2024-08-12
> > >
> > > Thank you! We appreciate your time and your constructive feedback.

---

### Official Review · Reviewer_JzeR · 2024-07-13

**Soundness:** 3
**Presentation:** 4
**Contribution:** 3
**Rating:** 6
**Confidence:** 5

**Summary:**

The paper proposes a pretrain-and-prompt-tuning paradigm to tackle the generalization challenge in RL. It pretrains a goal-conditioned transformer from task-agnostic datasets, and during fine-tuning, it constructs a goal sequence as a prompt and tunes that prompt via multi-arm bandit algorithms.

**Strengths:**

- The proposed pretrain-and-prompt-finetuning paradigm matches the trend of AI research and the future RL systems in real-world deployments.
- The conceptualization of the prompt and prompt-tuning in RL is interesting and promising.
- The proposed method is well formulated and presented with comprehensive experiments.

**Weaknesses:**

- The process of prompt optimization with multi-arm bandit modeling is not given in detail, and seems confusing and unconvincing.
- The relationship between the proposed method and the hierarchical RL/skill-based RL is not discussed and analyzed.
- The superiority over existing offline meta-RL methods is not elaborated in the text and not demonstrated in experiments.

**Questions:**

- The proposed method pretrains the “skills” using behavior cloning. If the dataset quality is not high enough, those skills could be sub-optimal temporal abstractions. Does this pretraining scheme further limits the performance, in the long term?
- The multi-arm bandit algorithm is employed to optimize the goal sequence, which is a NP-hard, combinatorial optimization problem. How to ensure it can find a good goal sequence using only a few iterations?
- In the multi-arm bandit modeling, what does one arm correspond to? It seems that one arm corresponds to an element from the prompt buffer. The prompt buffer expands as new prompts are added. The number of arms is infinite. The complexity of the multi-arm bandit algorithm could be high, and its theoretical guarantee can be hard to derive.
- At test time, the learner is given an unseen task with the “task goal”. Does this part need domain knowledge, to some extent? If the agent knows the task goal at any time, we can just train a goal-conditioned policy to do the job, as have been investigated in the literature.

**Limitations:**

- The multi-arm bandit modeling module is unclear with potentially high complexity and brittle theoretical guarantee.
- The task goal is needed at test time, which could further hinder its applicability to general RL problems.

---

> ### Author Rebuttal · Authors · 2024-08-07
>
> ## Thanks for your review! Here, we respond to your comments and address the issues. We hope to hear back from you if you have further questions!
>
> **Q1:** Using BC to train skills will limit the performance when the dataset quality is not high.
>
> **A1:** Indeed, the dataset quality impacts the performance of all imitation learning and offline RL methods. Our approach, however, does not rely solely on expert data but instead utilizes diverse, task-agnostic datasets containing meaningful behaviors, aligning with the literature on task-agnostic pre-training [1,2,3]. This enables the use of abundant data sources such as human gameplay videos for real-world tasks, without requiring expert labeling.
>
> In Appendix A.3, we present how dataset quality influences performance. Despite not using an expert policy for data collection, our method effectively composes short-term skills to solve long-horizon tasks like MazeRunner. Variations in dataset quality do impact model performance. In section 6, we discuss potential future directions to mitigate this, including the integration of offline RL and online finetuning methods.
>
> **Q2:** Questions about the MAB algorithm.
>
> **A2:** In our MAB framework, each arm corresponds to a prompt of goal sequence. Due to the combinatorial nature of potential prompts in the space $G^K$, this problem is NP-hard. However, we introduce specific insights to reduce the search space effectively, as detailed in Section 4.2:
>
> - Trajectory-based sampling: Instead of sampling prompts in the combinatorial space, we only sample prompts within the collected trajectories, thus grounding the search in feasible and relevant goal sequences.
> - Reward-guided exploration: We further refine the prompt search to only include those from the collected trajectories with best returns, enhancing the likelihood of improving performance.
> - Task-goal consistency: The final goal in each prompt remains fixed as the task goal, ensuring that exploration efforts are aligned with task completion.
>
> These strategies do not guarantee theoretical optimality due to the problem's complexity. Empirical results, as shown in Table 2 of our attached PDF, indicate that our method outperforms a vanilla MAB approach that utilizes the whole prompt space.
>
> We acknowledge the need for clearer exposition of our MAB framework and will revise the relevant sections in our manuscript.
>
> **Q3:** Does giving a task goal in test need domain knowledge?
>
> **A3:** No, domain knowledge is not required beyond knowing the task goal, which provides only partial information about the task's objective. The necessary subtasks to complete the task and the environment dynamics are unknown and vary across test tasks, making the challenge distinct from what a traditional goal-conditioned policy might handle. For instance:
> - In Crafter, the task goal is always to unlock all achievements, while the agent must adapt its policy to an unseen world map and identify an optimal sequence of subtasks.
> - In MazeRunner, the task goal is simply to reach a specified position $(x,y)$. The agent faces the challenge of navigating an unfamiliar maze layout to find the most efficient path.
>
> Our experiments show that the Goal-conditioned BC baseline, which is conditioned on the task goal, performs poorly across all tasks. This is because it lacks the ability to adapt to varying environmental dynamics in the test scenarios.
>
> **Q4:** The relationship to HRL/skill-based RL is not discussed.
>
> **A4:** Skill-based RL is closely related to our work, particularly in the context of offline skill pre-training for long-horizon tasks. As detailed in Section 1 (third paragraph) and Section 2 (first paragraph), we discuss relevant literature such as SPiRL[1], PTGM[3], and SKiMo[4], which focus on learning short-term skills offline and adapting to long-horizon tasks through online RL. These methods generally suffer from inefficiencies during online adaptation, which we address through prompt optimization.
>
> Hierarchical RL (HRL) also intersects with our study, as it commonly employs pre-trained skills for temporal abstraction in skill-based RL. Existing approaches [1,3,4] rely on HRL to learn a high-level policy during online adaptation. Unlike these methods, our approach sidesteps the inefficiencies associated with HRL by leveraging a pre-trained Transformer-based policy. This allows us to transform the online RL process into a more efficient prompt optimization process. We will add this discussion to our manuscript.
>
> **Q5:** Compare with existing offline meta-RL (OMRL) methods.
>
> **A5:** As outlined in Section 1 (second paragraph), OMRL and our approach target fundamentally different settings.
> -  OMRL necessitates task-specific training data with numerous trajectories per task, whereas our method leverages a **task-agnostic dataset** that captures diverse behaviors.
> - OMRL is typically confined to short-term tasks, enabling fast adaptation from suboptimal trajectories. In contrast, our focus is on **long-horizon tasks**, presenting a greater challenge that is distinct from the scope of OMRL.
>
> These differences justify our choice of finetuning and skill-based methods for baseline comparison, rather than OMRL approaches.
>
>
> [1] Pertsch et al., Accelerating reinforcement learning with learned skill priors, 2021.
>
> [2] Rosete-Beas et al., Latent plans for task-agnostic offline reinforcement learning, 2023.
>
> [3] Yuan et al., Pre-training goal-based models for sample-efficient reinforcement learning, 2024.
>
> [4] Shi et al., Skill-based model-based reinforcement learning, 2023.

---

> > ### Comment · Reviewer_JzeR · 2024-08-10
> >
> > Thank the authors for their detailed response, which has addressed most of my concerns. I will keep my score.

---

> > > ### Author Response · Authors · 2024-08-11
> > >
> > > Thanks again for your time reviewing our paper and for your constructive feedback!

---

### Official Review · Reviewer_V4S2 · 2024-07-30

**Soundness:** 3
**Presentation:** 3
**Contribution:** 3
**Rating:** 7
**Confidence:** 4

**Summary:**

This paper addresses the fast adaptation of pre-trained policy from task-agnostic datasets. The authors propose to avoid RL interactions on new tasks through the combination of Transformer-based policies to model multiple goals and efficient online adaptation through prompt optimization. The experiments demonstrate MGPO’s superior performance and efficient adaptation.

**Strengths:**

It is straightforward and reasonable to model the sequence of goals through prompt and  apply prompt optimization for online adaptation and goal learning in unseen environment. Converting goal switching capacity learning into fast online adaptation is natural.

The experimental results are sound and demonstrates good performance of the proposed method. The paper is well-written, and the visual examples well verify the interpretability of the method.

**Weaknesses:**

See questions.

**Questions:**

- What is the advantage of hindsight relabeling in this paper, as the environment already give feedback for goals. Why the hindsight relabeling is required? Could you please provide more discussion and explanations?

- How to determine the length of the sequences of goals while online adaptation. Is there any terminal signal or pre-defined length?

- While the prompt provide a way of fast online adaptation, I am curious about how advantage the transformer-based policy backbone provided, as there are already some RL works in fast adpation through optimizing a subset of network parameters, like optimizing the free parameter in [1]. How about the performance of this method if taking other neural network as the policy backbone, like cnn-based policies and lstm-based policies, where the prompt can be modeled as free parameters and learned through multi-task optimization, like [1]. Are prompt and transformer policy nessary for learning the goal switching?

[1] Huang B, Feng F, Lu C, et al. AdaRL: What, Where, and How to Adapt in Transfer Reinforcement Learning[C]//International Conference on Learning Representations.

- Could you please provide illustration of the learned prompts and the goals in the environment. How will the performance change if we permute the learned goals in prompt? If there is an obvious decrease, the proposed method indeed learn the sequetial relationship of the goals, right?


- Can you provide analysis about why the BC-finetune perform worse than Goal-conditioned BC?

- Some typos: like the bold formatting of the citation text on line 233.

**Limitations:**

See questions.

---

> ### Author Rebuttal · Authors · 2024-08-07
>
> ## Thanks for your review! Here, we respond to your comments and address the issues. We hope to hear back from you if you have further questions!
>
> **Q1:** Why the hindsight relabeling is required?
>
> **A1:** We use hindsight relabeling in the pre-training stage to learn from an offline dataset without environment interactions. Hindsight relabeling is essential for associating each trajectory with a specific sequence of goals, thereby training our policy to be goal-conditioned. Specifically, within each offline trajectory, we extract a sequence of goals that the agent was implicitly aiming to achieve during data collection. This approach allows the pre-trained policy to learn the relationship between sequences of goals and the corresponding actions, enhancing its ability to adapt to varied goal sequences and execute appropriate behaviors in different contexts.
>
>
> **Q2:** How to determine the goal sequence length during online adaptation.
>
> **A2:** In online adaptation, we maintain the maximal prompt length $K$ used during pre-training. The prompt length for each iteration is determined by sampling a number of subgoals $k \sim U[0, K-1]$ and sampling $k$ subgoals from the best trajectory collected. These details are provided in Section 4.1 and 4.2. We acknowledge and will correct a typo in line 199 regarding the sampling process, which should be $p' \sim P(p|\tau^*)$.
>
> **Q3:** Compare prompt optimization with the approach in AdaRL.
>
> **A3:** It's important to note that our work and AdaRL tackle distinct problems in fast adaptation. AdaRL focuses on **online meta-RL**, optimizing for fast adaptation to new tasks through multi-task online training. It leverages a compact context representation to facilitate few-shot adaptation primarily in **short-term tasks**, such as Cartpole and Atari Pong. Conversely, our approach focuses on **offline, task-agnostic** pre-training for **long-horizon** tasks. We aim at fast adaptation to new tasks with multiple subgoals or stages. To ensure a comprehensive evaluation, we included various baseline methods from this literature, including finetuning methods (BC-finetune) and skill-based methods (SPiRL and PTGM).
>
> **Q4:** Compare the Transformer policy to CNN-based and LSTM-based policies.
>
> **A4:** While CNNs and LSTMs are capable of sequence modeling, we choose the Transformer architecture due to its demonstrated superiority in offline RL [1,2] and imitation learning [3,4]. Its effectiveness in modeling trajectories surpasses other architectures, as evidenced by existing experiments [3,4]. This led us to adopt the Decision Transformer [1] architecture for both our method and all baseline comparisons.
>
> **Q5:** Illustrate the learned prompts and goals.
>
> **A5:** We maintain the order of sampled goals when constructing prompts through hindsight relabeling, ensuring the policy learns to achieve each goal sequentially. Permuting the goals within the prompt will cause the policy to attempt to reach subgoals in a different order. We illustrate our prompts and goals in the Kitchen environment in Figure 1 of our attached PDF. Permuting the goal sequence affects task execution, leading to failure in achieving the original task. This is critical for real-world tasks that require subtasks to be completed in a specific order.
>
>
> **Q6:**  Why does BC-finetune perform worse than Goal-conditioned BC?
>
> **A6:** BC-finetune requires substantial iterations to adapt to new tasks. Within the limited budget of 100 episodes in our experiment, it diverges from the original Goal-conditioned BC policy but fails to fully adapt, leading to underperformance. This is often due to the instability and initial performance decline of RL finetuning methods. To address this, we introduce an enhanced version, BC-finetune-KL, which integrates a KL-divergence loss with the initial policy to stabilize the finetuning process. As shown in Table 1 of the attached PDF, BC-finetune-KL shows some improvement over Goal-conditioned BC but still lags behind MGPO.
>
> **Q7:** Typos.
>
> **A7:** Thank you for pointing these out! We will fix the typos.
>
>
> [1] Chen et al., Decision Transformer: Reinforcement Learning via Sequence Modeling, 2021.
>
> [2] Li et al., A Survey on Transformers in Reinforcement Learning, 2023.
>
> [3] Dasari et al., Transformers for One-Shot Visual Imitation, 2020.
>
> [4] Kim et al., Transformer-based deep imitation learning for dual-arm robot manipulation, 2021.

---

> > ### Comment · Reviewer_V4S2 · 2024-08-11
> >
> > Thank you for your time. Most of my concerns have been addressed, and I will increase my rating.

---

> > > ### Author Response · Authors · 2024-08-12
> > >
> > > Thank you very much for raising the score! We appreciate your constructive feedback.

---

### Author Rebuttal · Authors · 2024-08-07

Thank you to all the reviewers for your insightful comments and constructive feedback. Here, we provide a summary of the reviews and our responses to the key points raised.

### Summary of positive feedback
- All reviewers: The proposed method (multi-goal pre-training and prompt optimization) is promising and exciting.
- Reviewer JzeR: the method is well formulated.
- Reviewer V4S2 and JzeR: the experimental results are sound and good.
- Reviewer V4S2: the paper is well-written, with good visual examples.

### Common questions addressed
- Comparison with meta-RL / offline meta-RL:  Meta-RL typically assumes access to multiple trajectories for each task and often tackles short-term tasks due to the challenges of few-shot task inference. Our study focuses on pre-training from **task-agnostic, offline** datasets and adapting to **long-horizon** tasks.
- Comparison with other skill-based RL methods: Online skill discovery methods, such as DIAYN [1], learn short-term skills via online RL. Our work aligns with the literature on **offline skill pre-training** [2,3,4] that acquires diverse skills from task-agnostic datasets and composes them for online task adaptation.

For individual issues, we have responded to each reviewer. We are committed to further refining our work based on your feedback. Some figures and tables are provided in the **attached PDF**. Please refer to it if needed.


[1] Eysenbach et al., Diversity is all you need: Learning skills without a reward function, 2018

[2] Pertsch et al., Accelerating reinforcement learning with learned skill priors, 2021.

[3] Yuan et al., Pre-training goal-based models for sample-efficient reinforcement learning, 2024.

[4] Shi et al., Skill-based model-based reinforcement learning, 2023.

---

> ### Author Response · Authors · 2024-08-11
>
> Dear Reviewers,
>
> Thank you for your insightful comments. We have addressed your initial questions through our rebuttal and are eager to clarify any further points you might raise. Please feel free to provide additional feedback. We greatly appreciate your continued engagement.
>
> Best regards,
>
> Authors

---

### Decision · Program_Chairs · 2024-09-25

**Decision:**

Accept (poster)

**Comment:**

The paper presents a method for efficient online adaptation and long-horizon tasks through pretraining transformers for predicting actions based on sequences of goals. The method is evaluated across a diverse set of tasks, demonstrating its practical impact.
However, there are some concerns regarding the presentation and technical explanations that could be improved. The authors are advised to enhance the clarity of their work by broadening the comparative analysis. This can be achieved by including a more detailed comparison with other task-agnostic pre-training methods to better position MGPO within the existing literature. Additionally, the authors should expand the discussion on the limitations of the prompt-based architecture, particularly addressing its volatility and unpredictability. Exploring potential solutions or future research directions that could mitigate these challenges would further strengthen the paper.